# Prevalence, severity, and risk factors of disability among adults living with HIV accessing routine outpatient HIV care in London, United Kingdom (UK): A cross-sectional self-report study

Darren A. Brown[1]*, Kelly K. O'Brien[2,3,4‡], Richard Harding[5,6‡], Philip M. Sedgwick[7], Mark Nelson[8,9], Marta Boffito[8], Agnieszka Lewko[10]

1 Therapies Department, Chelsea and Westminster Hospital NHS Foundation Trust, London, United Kingdom, 2 Department of Physical Therapy, University of Toronto, Toronto, Canada, 3 Institute of Health Policy, Management and Evaluation (IHPME), University of Toronto, Toronto, Canada, 4 Rehabilitation Sciences Institute (RSI), University of Toronto, Toronto, Canada, 5 Florence Nightingale Faculty of Nursing, Midwifery & Palliative Care, King's College London, London, United Kingdom, 6 Cicely Saunders Institute of Palliative Care, Policy and Rehabilitating, King's College London, London, United Kingdom, 7 Institute of Medical and Biomedical Education, St George's, University of London, London, United Kingdom, 8 Department of HIV Medicine, Chelsea and Westminster Hospital NHS Foundation Trust, London, United Kingdom, 9 Faculty of Medicine, Department of Medicine, Imperial College London, London, United Kingdom, 10 Centre for Allied Health, Kingston University and St George's University of London, London, United Kingdom

☯ These authors contributed equally to this work.
‡ KKO and RH also contributed equally to this work.
* Darren.Brown11@nhs.net

**Data Availability Statement:** Data are publicly available at the following links.

## Abstract

### Background

The study objectives were to measure disability prevalence and severity, and examine disability risk factors, among adults living with HIV in London, United Kingdom (UK).

### Methods

Self-reported questionnaires were administered: World Health Organization Disability Assessment Schedule 2.0 (WHODAS), HIV Disability Questionnaire (HDQ), Equality Act disability definition (EADD), and demographic questionnaire. We calculated proportion (95% Confidence Interval; CI) of "severe" and "moderate" disability measured using EADD and WHODAS scores ≥2 respectively. We measured disability severity with HDQ domain severity scores. We used demographic questionnaire responses to assess risk factors of "severe" and "moderate" disability using logistic regression analysis, and HDQ severity domain scores using linear regression analysis.

### Results

Of 201 participants, 176 (87.6%) identified as men, median age 47 years, and 194 (96.5%) virologically suppressed. Severe disability prevalence was 39.5% (n = 79/201), 95% CI

DOI: https://doi.org/10.5061/dryad.qnk98sfjn
Direct Link: https://datadryad.org/stash/
dataset/doi:10.5061%2Fdryad.qnk98sfjn.

**Funding:** This study/project is funded by the
National Institute for Health Research (NIHR)
Masters of Research (MRes) Programme. The
views expressed are those of the author(s) and not
necessarily those of the NIHR or the Department of
Health and Social Care. The funders had no role in
study design, data collection and analysis, decision
to publish, or preparation of the manuscript.

**Competing interests:** The authors have declared
that no competing interests exist.

[32.5%, 46.4%]. Moderate disability prevalence was 70.5% (n = 141/200), 95% CI [64.2%, 76.8%]. Uncertainty was the most severe HDQ disability domain. Late HIV diagnosis was a risk factor for severe disability [Odds Ratio (OR) 2.71; CI 1.25, 5.87]. Social determinants of health, economic inactivity [OR 2.79; CI 1.08, 7.21] and receiving benefits [OR 2.87; CI 1.05, 7.83], were risk factors for "severe" disability. Economic inactivity [OR 3.14; CI 1.00, 9.98] was a risk factor for "moderate" disability. Economic inactivity, receiving benefits, and having no fixed abode were risk factors ($P \leq 0.05$) for higher HDQ severity scores in physical, mental and emotional, difficulty with day-to-day activities, and challenges to social participation domains. Personal factors, identifying as a woman and being aged <50 years, were risk factors ($P \leq 0.05$) for higher HDQ severity scores in mental and emotional, uncertainty, and challenges with social participation domains.

### Conclusions

People living with well-controlled HIV in London UK experienced multi-dimensional and episodic disability. Results help to better understand the prevalence, severity, and risk factors of disability experienced by adults living with HIV, identify areas to target interventions, and optimise health and functioning.

## Background

HIV is now considered a chronic [1] and episodic [2, 3] health condition. For the 37.9 million people living with HIV globally [4], universal access to antiretroviral therapies offer normal life expectancy [5]. With access to effective and tolerable antiretroviral therapies, the number of people living with HIV aged 50 years or older are increasing at exponential and unprecedented rates [6]. It is estimated that globally, 7.5 million people living with HIV are aged 50 years or older [7]. Furthermore, by 2028 over half of people living with HIV In the United Kingdom (UK) will be aged $\geq$50 years [8], with similar patterns observed elsewhere in Europe and North America [9]. As people live longer with chronic HIV, they are susceptible to health conditions arising from the underlying infection, potential side effects of treatments, and ageing [10], resulting in increasingly more prevalent multi-morbidity [1, 11]. Common concurrent health conditions include cardiovascular disease [12], diabetes [13], bone and joint disorders [14, 15], neurocognitive disorders [16, 17], chronic pain [18], mental health conditions [19], cancer [20], and frailty [21]. People living with HIV can also experience additional challenges of stigma, ageism, income insecurity, and lack of social support, which may impact or intersect with issues of living and ageing with HIV [22–25]. Collectively the physical, mental and social health challenges experienced can be conceptualised as disability [2]. Many people living and ageing with HIV on long-term antiretroviral therapy now face new or worsening experiences of a wide variety of disability [10, 26], with gender, physical symptoms, depression, antiretroviral therapy adherence, and duration living with HIV associated with experiencing disability [27–29].

The conceptualisation of disability is complex and has evolved over time [30]. The predominating framework of disability, the "*International Classification of Functioning, Disability and Health*" (ICF) [31], combines elements of both medical [32] and social models [33] of disability, leading to a "*bio-psycho-social*" framework. The ICF describes disability as multidimensional and the outcome of interactions between a person's health condition(s) and context

(environmental and personal factors), involving one or more dysfunctions at the level of impairments, activity limitations and participation restrictions. The ICF is not specific to any health condition and may not accurately capture the complexity of HIV [2]. The "*Episodic Disability Framework*" (EDF) presents a new way to conceptualise disability based on the experience of people living with HIV [2]. It conceptualises disability as multi-dimensional and episodic, describing the health-related consequences of HIV, adverse effects of treatments, and concurrent health conditions that may fluctuate over time [3]. Neither conceptualisation is inherently better than the other. Disability can however be broadly defined as any physical, cognitive, mental or emotional impairments, difficulty with day-to-day activities, challenges to social inclusion, or uncertainty, that can be episodic in nature, whereby disability reflects the interaction between a person's body and the society in which they live [2, 34].

Measuring disability is a critical component of care as people live longer with HIV [35] to determine the prevalence and impact of disability [36], prevent discrimination, monitor inequalities, identify service needs and address barriers to equal participation in society [37]. Measuring disability prevalence poses several challenges, not least since there is a wide range of definitions, measurement tools, and purposes for data collection [38]. Measuring impairments alone is not an adequate proxy for disability, since people with the same impairment can experience different types and degrees of restriction [39]. The majority of studies estimating disability prevalence among people living with HIV have focused only on measurements of single impairments [40]. Tools have been developed to standardise disability measurement reflecting the multi-dimensional nature of disability, including generic tools such as the Washington Group on Disability Statistics [41], Equality Act disability definition (EADD) of Great Britain [42], World Health Organization Disability Assessment Schedule 2.0 (WHODAS) [43], and Model Disability Survey [44], alongside condition specific tools such as the HIV Disability Questionnaire (HDQ) [45]. In the UK general population, disability prevalence is estimated with EADD [42, 46] to meet the needs of government policy and equalities monitoring [37]. There is no known literature reporting disability prevalence among people living with HIV in the UK, using this generic approach to disability measurement within populations.

With the variety of measurement approaches available, each with benefits and limitations, a single measure to estimate disability prevalence might be problematic [38]. It has been proposed that at least two disability prevalence rates should be reported, to represent a range of disability including a moderate threshold and a more severe threshold [38]. To our knowledge there is no known evidence estimating disability prevalence representing a range of disability severity, nor an understanding of potential risk factors for disability among people living with HIV in the UK.

The objectives of our study were to estimate the prevalence and severity of disability, and examine the potential risk factors of disability among adults living with HIV accessing routine in-person outpatient HIV care in London, UK.

## Methods

We conducted a quantitative cross-sectional study, using self-report questionnaires, to measure prevalence and severity of disability and examine potential risk factors of disability, among adults living with HIV accessing routine in-person outpatient HIV care in London, UK. Quality criteria for reporting cross-sectional studies set out in the STROBE statement were used to guide our methodological approach [47].

### Patient and public involvement

Patients and the public were involved throughout the research study design and management. Sources of Patient and Public Involvement (PPI) included host organisation "patient champion",

host organisation directorate peer review, and Positively UK; a UK based charity for people living with HIV.

## Ethics

We obtained ethical approval from NHS Research Ethics Committee and Health Research Authority (REC reference 18/LO/0590; IRAS 236835). All participants provided written informed consent to participate in the study.

## Participants

We recruited adults ≥18 years living with HIV and on antiretroviral HIV treatment for ≥6 months attending routine in-person outpatient HIV care from three outpatient HIV clinics in central London, UK, between May-July 2018. We used convenience sampling, approaching alternative patients in clinic requesting participation in the study.

## Sample size

No prerequisite data exists on prevalence of disability among the estimated 105,200 people living with HIV in the UK [48]. An exploratory sample size of 200 participants was proposed.

## Data collection

Our primary outcome was disability. We measured disability using three approaches, by administering the following three self-completed paper-based questionnaires after routine in-person outpatient HIV care at the clinic visit; (a) demographic and HIV questionnaire including EADD, (b) WHODAS, (c) HDQ.

**Measuring disability.**

1. **EADD**: The EADD [42] was developed through a programme of technical development and public consultation following a cross-government Equality Data review in 2007. The EADD tries to reflect the definitions that appeared in legal terms in the Disability Discrimination Act 1995 and the subsequent Equality Act 2010. According to the Equality Act a person is disabled if they have a physical or mental impairment and the impairment has a substantial and long-term adverse effect on their ability to carry out normal day-to-day activities [37]. The classification questions making up the EADD are: (a) "*do you have any physical or mental health conditions or illnesses lasting or expecting to last 12-months or more*?"; (b) "*do any of your conditions of illnesses reduce your ability to carry out day-to-day activities*?" [42]. A person is counted as disabled if they answer "*yes*" to both classification questions.

2. **WHODAS 2.0 (12 Item)**: The WHODAS [43] is a self-administered generic measurement tool of functioning and disability applicable across cultures in adult populations, and directly linked to the ICF [43, 49]. It measures an individual's difficulty in performing specific functions over the previous 30 days across six disability domains: i) cognition, ii) mobility, iii) self-care, iv) getting along, v) life activities and vi) participation). When completing the WHODAS individuals provide an answer for each question on a 5-point Likert scale (range 0–4) with higher scores indicating increasing difficulty completing the task. The WHODAS provides "simple" and "complex" sum scores. In "simple" scoring, the scores assigned to each item are summed (range 0–48) with higher scores suggestive of greater disability [49]. In "complex" or item response theory-based scoring, multiple levels of difficulty are factored for each item using downloadable scoring sheet from the

WHODAS website, providing a disability range from 0 (no disability) to 100 (total disability) [49]. The WHODAS has been used among adults living with HIV in high [29, 50–52], middle-, and low-income countries [28, 29, 53–57]. The WHODAS has high internal consistency and test-retest reliability [58, 59], plus rigorous validity and cross-cultural testing spanning 19 countries [49]. The WHODAS is also validated in patients with chronic diseases [60] and people living with HIV [56]. The short-form WHODAS (12-item) used in our study explains 81% variance of long-form (36-item), and average short-form administration time is 5 minutes [43, 49]. For the purpose of disability statistics, categorisation thresholds have been developed based on WHODAS scores, to identify people living with HIV experiencing disability (score ≥2, representing at least two mild/moderate or one moderate/severe limitation) [27, 28], and any level of functional limitation (score ≥1, representing at least one mild/moderate limitation) [29, 52]. These thresholds permit prevalence estimates of disability and functional limitation among people living with HIV in South Africa [27–29] and United States [29, 52].

3. **HDQ**: The HDQ is a 69-item self-administered questionnaire developed from the EDF, through a community-academic partnership, to describe the presence, severity and episodic nature of disability experienced by people living with HIV [2, 45]. The HDQ includes six disability domains: i) physical, ii) cognitive and, iii) mental and emotional health symptoms and impairments, iv) uncertainty, v) difficulty with day-to-day activities, and vi) challenges to social inclusion, and one 'good day/bad day' health classification item. Participants are asked to rate the level of presence and severity of each health challenge on a given day ranging from 0 (not at all) to 4 (extreme). HDQ presence, severity and episodic scores are linearly transformed to a score ranging from 0 to 100, with higher scores indicating a greater presence, severity and episodic nature of disability [61]. The HDQ has demonstrated sensibility, validity, internal consistency reliability, test-retest reliability, and varied precision of measurement in samples of adults living with HIV in Canada, Ireland, United States, and UK [35, 51, 61–63]. Median administration time is 8–15 minutes [51].

**Demographic and HIV characteristics.** We administered the paper-based self-report demographic and HIV questionnaire (S1 File) which included 19 items capturing: age (years), gender identity, sexual orientation, ethnicity, number of years since HIV diagnosis, whether HIV was diagnosed "late" (CD4 <350 cell/mm$^3$ at diagnosis), antiretroviral therapy use, most recent viral load (copies/ml), employment status, housing situation, use of adaptations to support day-to-day activities, educational attainment, transport, diagnosed concurrent health conditions, health status, receiving or providing care, receipt of benefits, receiving rehabilitation in past 12-months, and EADD [42, 46]. Clinical characteristics were collected from electronic records only when participants could not provide data including number of years since HIV diagnosis, most recent viral load, and diagnosed concurrent health conditions.

## Data analysis

**Participant demographic, HIV, and disability characteristics.** For continuous variables, we calculated median, Lower Quartile (LQ) and Upper Quartile (UQ), and range including: age, number of years since HIV diagnosis, number of concurrent health conditions, number of transportation modes, and number of rehabilitation professionals received care from in the past 12-months.

For categorical variables, we reported frequencies and percentages including: gender identity, sexual orientation, ethnicity, whether HIV was diagnosed "late" (CD4 <350 cell/mm$^3$ at

diagnosis), antiretroviral therapy use, viral load <50 copies/ml, employment status, housing situation, use of adaptations to support day-to-day activities, educational attainment, transport, health status, receiving or providing care, receipt of benefits, and received rehabilitation in past 12-months.

We reported disability characteristics as measured by domain scores of the WHODAS and HDQ. We reported frequency and percentages for WHODAS simple scores in response to all 12-items, presence of any functional limitation (score ≥1) per WHODAS domain, WHODAS total number of limitations, and WHODAS difficulty levels. WHODAS "simple" and "complex" sum scores were both reported to follow recommendations to report "complex" sum score [43], and take account of current debate promoting "simple" sum score [64]. Both "simple" and "complex" sum scores were presented as mean and standard deviation (SD), plus median and 25-75th percentile, to align with normative data presented as mean and SD [58] and existing WHODAS 12-item literature reporting either mean and SD, or median and LQ-UQ [65]. We reported median, LQ, UQ, and range for HDQ presence, severity, and episodic total scores and scores per HDQ domain.

**Disability prevalence.** The World Bank recommends estimating disability prevalence with at least two rates representing a range of severity [38]. For the purpose of this study we reported two disability prevalence rates representing "severe" and "moderate" thresholds:

1. Severe disability: We defined severe disability with the EADD, which is used for UK disability national surveillance [66]. This census-based approach may correspond to people with most severe disability [38]. Participants were defined as experiencing "severe" disability if they self-rated "yes" to both EADD classification questions.

2. Moderate disability: We defined moderate disability with the WHODAS, the only known approach estimating disability prevalence among people living with HIV [27–29, 52]. This survey-based approach may correspond to people with moderate disability [27]. Participants were defined as experiencing "moderate" disability with WHODAS scores ≥2, representing at least two mild/moderate or one moderate/severe limitation on WHODAS items [27, 28].

We reported disability prevalence representing both severe and moderate thresholds, as frequency, percentage, and 95% confidence interval (CI).

**Disability severity.** We calculated median, LQ and UQ of each of the six HDQ domain severity scores. Interpretability of HDQ scores among people living with HIV remains unknown hence we describe severity on the continuous scale (0–100) whereby higher scores indicated greater severity of disability [35].

**Potential risk factors of disability.** We examined potential risk factors of disability using logistic and linear regression analysis for categorical and continuous variables respectively. Our outcome variable (disability) was defined using three approaches: (a) severe disability (EADD); (b) moderate disability (WHODAS score ≥2); (c) all six HDQ domain severity scores.

The following variables were considered as potential risk factors in the models:

- *Age*: "<50 years "and "≥50 years", because ≥50 years is the most used definition of "older" within current HIV literature [67, 68], and British HIV Association standards of care [69].

- *Gender*: Where gender breakdowns were presented, Transwomen and Transmen were included in the gender groups with which they self-identified in accordance with Public Health England data on people living with HIV [70], resulting in categorised "man" and "woman".

- *Sexual orientation*: "Heterosexual" and "Lesbian/Gay/Bisexual/Other" in accordance with UK Government Statistical Service (GSS) harmonised principles [71].

- *Ethnicity*: "White" and "Black, Asian, and Minority Ethnicities (BAME)" in accordance with UK GSS harmonised principles [72].

- *Late HIV diagnosis*: Whether HIV was diagnosed "late" was defined by CD4 count <350 cell/mm$^3$ at diagnosis, resulting in categorised "CD4 count <350 cell/mm$^3$ at diagnosis" and "CD4 count >350 cell/mm$^3$ at diagnosis".

- *Employment*: "economically active" and "economically inactive", based on the Labour Force Survey whereby economic inactivity is defined by people not in employment who have not been seeking work within the last 4 weeks and/or are unable to start work within the next 2 weeks [73].

- *Housing situation*: "owner occupied/privately rented/social rented" and "no fixed abode/ other", based on UK GSS harmonised principles [74].

- *Educational attainment*: categorised by achieving educational qualifications at "degree level or above" or "any other kind of qualification" according to UK GSS harmonised principles [75].

- *Care and Support*: "does not receive or provide care" and those who "receive or provide care", whereby receiving care includes social services care and informal/unpaid care [76].

- *Benefits*: "not in receipt of benefits" and "in receipt of benefits" whereby being in receipt of benefits includes receiving working age benefits, disability benefits, pensioner benefits, child benefits, social fund, and/or other benefits, in accordance with UK GSS harmonised principles [77].

- *Rehabilitation*: "received rehabilitation in past 12-months" and "no rehabilitation in past 12-months" whereby rehabilitation is defined as receiving Physiotherapy, Occupational Therapy, Speech and Language Therapy, and/or complementary and alternative services [78].

We conducted logistic regression analysis to determine risk factors for severe (EADD) and moderate (WHODAS score ≥2) disability. We calculated unadjusted and adjusted odds ratios (OR) for each risk factor. We conducted multivariate regression (generalised liner model) for HDQ domain severity scores. We estimated marginal means, mean difference, 95% CI, and *P*-values for each risk factor as a potential predictor of disability.

Level of significance was ≤0.05 (5%). We did not adjust for multiple hypothesis testing, as adjustment using Bonferroni's correction factor tends to be too conservative and result in an excess of Type II errors [79]. We used a pragmatic approach, with greater emphasis given to statistical significance below 0.01 (1%). Data analysis was performed using SPSS Version 25 [80].

## Results

A total of 316 potential participants were screened for inclusion, with n = 204 (64.6%) consenting to participate, and n = 3 not meeting eligibility criteria of antiretroviral therapy use for ≥6 months. The final sample was n = 201, with one person not completing all three questionnaires. Median time to complete the questionnaires was 22 minutes (range 9–54 minutes).

### Participant characteristics

**Sample demographic and HIV characteristics.**  Participant characteristics are shown in Table 1. The majority identified as men (87.6%), of gay/lesbian sexual orientation (78.6%), and

**Table 1. Participant characteristics.**

| Age | Median (LQ, UQ) [Range] |
|---|---|
| Median Age | 47 (37, 56) [22–88] |
| Number of participants aged ≥50 years; number (%) | 82 (41.0) |
| **Gender Identity** | **Number (%)** |
| Man / Boy | 176 (87.6) |
| Woman / Girl | 20 (10.0) |
| Transwomen / Transgirl | 5 (2.4) |
| **Sexual Orientation** | **Number (%)** |
| Gay / Lesbian | 158 (78.6) |
| Heterosexual / Straight | 31 (15.4) |
| Bisexual | 8 (4.0) |
| Don't know / Prefer not to say / Other | 4 (2.0) |
| **Ethnicity** | **Number (%)** |
| white—English / Welsh / Scottish / Northern Irish / British | 93 (46.3) |
| white—Irish | 6 (3.0) |
| white—any other white background | 47 (23.4) |
| Mixed / Multiple Ethnic groups—white and Black Caribbean | 4 (2.0) |
| Mixed / Multiple Ethnic groups—white and Black African | 5 (2.5) |
| Mixed / Multiple Ethnic groups—white and Asian | 1 (0.5) |
| Mixed / Multiple Ethnic groups—any other mixed / multiple Ethnic background | 4 (2.0) |
| Asian / Asian British—Indian | 6 (3.0) |
| Asian / Asian British—Pakistani | 1 (0.5) |
| Asian / Asian British—Chinese | 1 (0.5) |
| Asian / Asian British—any other Asian background | 2 (1.0) |
| Black / Black British—African | 22 (10.9) |
| Black / Black British—Caribbean | 4 (2.0) |
| Black / Black British—any other Black / African / Caribbean background | 1 (0.5) |
| Other Ethnic group—Arab | 2 (1.0) |
| Other Ethnic group—any other Ethnic group | 2 (1.0) |
| **Number of years since HIV diagnosis** | **Median (LQ, UQ) [Range]** |
| Median years since HIV diagnosis | 11 (5, 21) [1–37] |
| **HIV diagnosed late** | **Number (%)** |
| CD4 count <350 cell/mm$^3$ at diagnosis | 104 (51.7) |
| **Taking Antiretroviral Therapy** | **Number (%)** |
| Yes | 201 (100.0) |
| **Viral Load Undetectable** | **Number (%)** |
| Viral load <50 copies/ml | 194 (96.5) |
| **Concurrent health conditions** | **Median (LQ, UQ) [Range]** |
| Median number of conditions in addition to living with HIV | 2 (1,5) [0–18] |
| **Self-rated general health status** | **Number (%)** |
| Very good | 78 (38.8) |
| Good | 74 (36.8) |
| Fair | 35 (17.4) |
| Poor | 14 (7.0) |
| **Employment** | **Number (%)** |
| Self-employed | 42 (20.9) |
| Full-time employed | 70 (39.3) |
| Part-time employed | 13 (6.5) |

(*Continued*)

**Table 1.** (Continued)

| | |
|---|---|
| Not working; available to start work in 2 weeks | 4 (2.0) |
| Not working; looked for work in past 4 weeks | 3 (1.5) |
| Waiting to start a new job | 2 (1.0) |
| Unemployed | 14 (7.0) |
| Retired | 20 (10.0) |
| Full time student / at school | 4 (2.0) |
| Long term sick or disabled | 20 (10.0) |
| **Accommodation** | **Number (%)** |
| Owner occupied | 75 (37.3) |
| Privately rented accommodation | 73 (36.3) |
| Social rented housing | 46 (22.9) |
| No fixed abode | 3 (1.5) |
| Other | 4 (2.0) |
| **Household information** | **Number (%)** |
| Lives alone | 89 (44.3) |
| Lives with friends | 32 (15.9) |
| Lives with family | 36 (17.9) |
| Lives with children | 6 (3.0) |
| Other | 38 (18.9) |
| **Housing adaptations to support day-today activities** | **Number (%)** |
| No | 171 (85.1) |
| **Educational Attainment** | **Number (%)** |
| Attained educational qualifications | 185 (92.0) |
| Attained professional / vocational qualifications | 133 (66.2) |
| **Highest educational qualification** | **Number (%)** |
| Degree level or above | 117 (58.2) |
| Any other kind of qualification | 73 (36.3) |
| No qualifications | 11 (5.5) |
| **Modes of transportation to appointment** | **Number (%)** |
| Underground (tube) | 83 (41.3) |
| Bus | 60 (29.9) |
| Walk | 49 (24.4) |
| Train | 42 (20.9) |
| Car | 22 (10.9) |
| Bike | 7 (3.5) |
| Other | 9 (4.5) |
| **Number of transport modes** | **Median (LQ, UQ) [Range]** |
| Median number of transportation modes | 1 (1,2) [1–4] |
| **Care and support** | **Number (%)** |
| Receive care from social services | 10 (5.0) |
| Receive informal / unpaid care | 19 (9.5) |
| Provide care for others (eg: friends or family) | 29 (14.4) |
| Do not receive of provide care | 145 (72.1) |
| **Benefits and tax credits** | **Number (%)** |
| Working age benefits | 36 (17.9) |
| Disability benefits | 36 (17.9) |
| Child benefits | 8 (4.0) |
| Pensioner benefits | 10 (5.0) |

(*Continued*)

**Table 1.** (Continued)

| | |
|---|---|
| Social fund | 1 (0.5) |
| Other benefits | 1 (0.5) |
| None | 141 (70.1) |
| **Received care from rehabilitation professionals in past 12-months** | **Number (%)** |
| Physiotherapy | 51 (25.4) |
| Occupational Therapy | 15 (7.5) |
| Speech and Language Therapy | 2 (1.0) |
| Complimentary and alternative services | 29 (14.4) |
| None | 130 (64.7) |
| **Number of rehabilitation professionals accessed in past 12-months** | **Median (LQ, UQ) [Range]** |
| Median number of rehabilitation professionals | 0 (0, 1) [0–3] |

white ethnicity (72.7%). The median age was 47 years, with 41.0% aged $\geq$50 years. The sample were mostly living with well controlled HIV with all participants taking antiretroviral therapy with 96.5% achieving HIV viral suppression. Median number of years since HIV diagnosis was 11 years (25-75th percentile: 5–21). Half were diagnosed with HIV late (51.7%). Participants reported living with a median of two concurrent health conditions in addition to HIV, with most self-reporting general health status "very good" or "good". Participants were mostly economically active, living in owned or private/social rented accommodation, living alone, and half achieving degree level qualifications. The majority did not receive or provide care, receive benefits/tax credits, or receive rehabilitation in past 12-months.

**Sample disability characteristics.** The WHODAS simple scores in response to all 12-items are shown in S1 Table. Frequency of any functional limitation (score $\geq$1) within each of the six WHODAS disability domains were; challenges to social participation (n = 208, 52.0%), challenges getting along (n = 150, 37.5%), challenges to life activities (n = 166, 41.5%), cognitive health challenges (n = 147, 36.7%), mobility challenges (n = 145, 36.2%), and challenges with self-care (n = 90, 22.5%). Any level of functional limitation (WHODAS score $\geq$1) was reported by n = 159 (79.5%), whereby n = 26 (13.0%) scored one limitation, n = 18 (9.0) two limitations, n = 14 (7.0%) three limitations, n = 101 (50.5%) four or more limitations, and n = 21 (10.5%) scoring all twelve limitations. Difficulty levels across all WHODAS items were "mild difficulty" n = 333 (38.8%), "moderate difficulty" n = 305 (33.7%), "severe difficulty" n = 143 (15.8%), and "extreme difficulty/cannot do" n = 125 (13.8%). WHODAS "simple" sum score were mean 9.4 (SD; 11.3), median 5.0 (25-75th percentile; 1.0–12.0). WHODAS "complex" sum score were mean 19.6 (SD; 23.7), median 10.4 (LQ-UQ; 2.1–25.6). The HDQ total and domain scores are shown in Table 2. The most present, severe, and episodic HDQ disability domains were "uncertainty", "uncertainty", and "physical symptoms and impairments" respectively, with n = 177 (88.5%) reported completing the HDQ on a good day.

## Disability prevalence

The estimated prevalence of severe disability was 39.5% (n = 79), 95% CI [32.5%, 46.4%]. In total 102 (50.7%) respondents reported presence of any physical or mental health conditions or illnesses lasting or expecting to last 12-months or more, whilst 35 (34.3%) reported subsequent activity restrictions "Yes, a lot", and 44 (43.1%) "Yes, a little".

The estimated prevalence of moderate disability was 70.5%, 95% CI [64.2%, 76.8%], with 141 respondents scoring $\geq$2 on WHODAS.

**Table 2. Median HDQ domain scores.**

| HDQ domain (# items) | HDQ Presence Score: Median (LQ, UQ) [Range] | HDQ Severity Score: Median (LQ, UQ) [Range] | HDQ Episodic Score: Median (LQ, UQ) [Range] |
|---|---|---|---|
| Physical symptoms and impairments | 30.0 (15.0, 53.8) | 11.3 (5.0, 26.3) | **20.0 (10.0, 40.0)** |
| (20 items) | [0.0–100.0] | [0.0–83.8] | **[0.0–100.0]** |
| Cognitive symptoms and impairments | 33.3 (0.0, 100.0) | 8.3 (0.0, 25.0) | 0.0 (0.0, 66.7) |
| (3 items) | [0.0–100.0] | [0.0–100.0] | [0.0–100.0] |
| Mental and emotional health symptoms and impairments | 45.5 (18.2, 72.7) | 13.6 (4.5, 34.1) | 18.2 (0.0, 54.5) |
| (11 items) | [0.0–100.0] | [0.0–100.0] | [0.0–100.0] |
| Uncertainty or worry about the future | **57.1 (28.6, 78.6)** | **23.2 (10.7, 38.9)** | 7.1 (0.0, 42.9) |
| (14 items) | **[0.0–100.0]** | **[0.0–100.0]** | [0.0–92.9] |
| Difficulties with day-to-day activities | 11.1 (0.0, 44.4) | 2.8 (0.0, 6.7) | 0.0 (0.0, 22.2) |
| (9 items) | [0.0–100.0] | [0.0–80.6] | [0.0–100.0] |
| Challenges to taking part in social and community life | 33.3 (8.3, 58.3) | 14.6 (4.2, 29.2) | 0.0 (0.0, 25.0) |
| (12 items) | [0.0–100.0] | [0.0–100.0] | [0.0–83.3] |
| Total HDQ Score | 36.2 (21.7, 59.4) | 13.4 (6.3, 28.8) | 17.4 (5.8, 36.2) |
| | [0.0–98.6] | [0.0–86.6] | [0.0–97.8] |

Higher scores indicate greater presence, severity and episodic nature of disability.

**Bold** indicates the highest scores across all domains.

## Disability severity

The HDQ domain severity scores are shown in Table 2. The three highest severity scores were disability domains: uncertainty or worry about the future (median; LQ, UQ: 23.2; 10.7, 38.9); challenges to taking part in social and community life (14.6; 4.2, 29.2); mental and emotional health symptoms and impairments (13.6; 4.5, 34.1).

## Potential risk factors of disability

**Risk factors of severe disability.** Participants who were diagnosed with HIV late (OR 2.71; 95% CI: 1.25, 5.87), were economically inactive (OR 2.79; 95% CI: 1.08, 7.21), received benefits (OR 2.87; 95% CI: 1.05, 7.83), and received rehabilitation in past 12-months (OR 4.56; 95% CI: 2.11, 9.86) were associated with statistically significant increased odds for "severe" disability.

**Risk factors of moderate disability.** Participants who were economically inactive (OR 3.14; 95% CI: 1.00, 9.89), and received rehabilitation in past 12-months (OR 3.41; 95% CI: 1.44, 8.10) were associated with statistically significant increased odds for "moderate" disability. All participants categorised as no fixed abode met threshold for moderate disability, therefore associations could not be analysed. Potential risk factors as predictors severe and moderate disability are shown in Table 3.

**Risk factors of HDQ domains.** Physical symptoms and impairments mean severity HDQ domain scores were higher among participants who identified as women (95% CI: 1.22, 18.41; $P = 0.025$), were economically inactive (4.51, 15.54; $P<0.001$), of no fixed abode (3.13, 24.86; $P = 0.012$), received benefits (1.88, 13.52; $P = 0.010$), and received rehabilitation in past 12-months (30.20, 42.89; $P<0.001$).

Cognitive symptoms and impairments mean severity HDQ domain scores were higher among participants who were economically inactive (4.88, 21.91; $P = 0.002$), received benefits (3.73, 21.72; $P = 0.006$), and received rehabilitation (29.45, 49.05; $P = 0.007$) in past 12-months.

**Table 3. Potential risk factors as predictors of severe or moderate disability.**

| Characteristic | Severe disability | | Moderate disability | |
| --- | --- | --- | --- | --- |
| | (EADD) | | (WHODAS score ≥2) | |
| | Unadjusted Odds Ratio (95% CI) | Adjusted Odds Ratio (95% CI) | Unadjusted Odds Ratio (95% CI) | Adjusted Odds Ratio (95% CI) |
| **Age** | | | | |
| < 50 years | (1) | (1) | (1) | (1) |
| ≥ 50 years | **3.33 (1.84, 6.02)** | 1.02 (0.43, 2.39) | 1.25 (0.67, 2.33) | 0.43 (0.18, 1.03) |
| **Gender Identity** | | | | |
| Man | (1) | (1) | (1) | (1) |
| Woman | 1.50 (0.65, 3.48) | 1.72 (0.34, 8.83) | **5.56 (1.27, 24.38)** | 1.55 (0.22, 10.98) |
| **Sexual Orientation** | | | | |
| Heterosexual | (1) | (1) | (1) | (1) |
| Lesbian/Gay/Bisexual/Other | 0.55 (0.26, 1.19) | 1.24 (0.26, 6.01) | **0.14 (0.03, 0.59)** | 0.13 (0.01, 1.43) |
| **Ethnicity** | | | | |
| White | (1) | (1) | (1) | (1) |
| BAME | 0.76 (0.40, 1.44) | 0.38 (0.14, 1.03) | 1.72 (0.83, 3.56) | 0.92 (0.38, 2.25) |
| **Late HIV Diagnosis** | | | | |
| CD4 count ≥350 cell/mm$^3$ at diagnosis | (1) | (1) | (1) | (1) |
| CD4 count <350 cell/mm$^3$ at diagnosis | **3.75 (2.05, 6.88)** | **2.71 (1.25, 5.87)** | **1.91 (1.03, 3.55)** | 1.31 (0.61, 2.79) |
| **Employment** | | | | |
| Economically Active | (1) | (1) | (1) | (1) |
| Economic Inactivity | **6.56 (3.36, 12.79)** | **2.79 (1.08, 7.21)** | **4.47 (1.89, 10.57)** | **3.14 (1.00, 9.89)** |
| **Housing Situation** | | | | |
| Owner/Private Rent/Social Rent | (1) | (1) | (1) | (1) |
| No fixed abode / other | 4.05 (0.77, 21.43) | 2.26 (0.32, 16.04) | - | - |
| **Educational Attainment** | | | | |
| Degree level or above | (1) | (1) | (1) | (1) |
| Any other kind of qualification | 1.36 (0.74, 2.48) | 0.89 (0.41, 1.92) | 1.81 (0.92, 3.56) | 1.62 (0.75, 3.49) |
| **Care and Support** | | | | |
| Does not receive or provide care | (1) | (1) | (1) | (1) |
| Receives or provides care | **3.43 (1.81, 6.52)** | 1.19 (0.46, 3.09) | **3.96 (1.67, 9.37)** | 1.79 (0.61, 5.25) |
| **Benefits** | | | | |
| Not in receipt of benefits | (1) | (1) | (1) | (1) |
| Receives benefits | **6.94 (3.53, 13.67)** | **2.87 (1.05, 7.83)** | **4.34 (1.84, 10.26)** | 1.38 (0.41, 4.67) |
| **Rehabilitation** | | | | |
| No rehabilitation in past 12 months | (1) | (1) | (1) | (1) |
| Received rehabilitation in past 12 months | **5.63 (2.99, 10.60)** | **4.56 (2.11, 9.86)** | **3.63 (1.70, 7.74)** | **3.41 (1.44, 8.10)** |

**Bold** indicates 95% CI does not cross unity, and therefore OR statistically significantly different from unity (P≤0.05).

Mental and emotional health symptoms and impairments mean severity HDQ domain scores were higher among participants aged <50 years (-14.09, -1.23; $P = 0.020$), who identified as women (1.85, 25.72; $P = 0.024$), were economically inactive (7.76, 23.07; $P<0.001$), of no fixed abode (7.37, 37.54; $P = 0.004$), and received benefits (0.19, 16.36; $P = 0.045$).

Uncertainty mean severity HDQ domain scores were higher among participants aged <50 years (-13.41, -0.79; $P = 0.027$), who identified as women (8.52, 31.96; $P = 0.001$), were

economically inactive (0.25, 15.29; *P* = 0.043), of no fixed abode (2.26, 31.89; *P* = 0.024), and received rehabilitation the past 12-months (45.82, 63.12; *P* = 0.004).

Difficulties with day-to-day activities mean severity HDQ domain scores were higher among participants who were economically inactive (9.92, 22.01; *P*<0.001), of no fixed abode (0.02, 23.86; *P* = 0.050), received benefits (4.19,16.97; *P* = 0.001), and received rehabilitation the past 12-months (28.65, 42.57; *P*<0.001).

Challenges to taking part in social and community life mean severity HDQ domain scores were higher among participants who were aged < 50 years (-11.08, -0.13; *P* = 0.045), identified as women (4.28, 24.62; *P* = 0.005), were economically inactive (6.25, 19.30; *P*<0.001), of no fixed abode (3.23, 28.95; *P* = 0.014), received benefits (1.63, 15.41; *P* = 0.015), and received rehabilitation in the past 12-months (35.47, 50.49; *P* = 0.001). Multivariate associations between potential risk factors and mean HDQ domain severity scores are shown in Table 4.

## Discussion

Disability was experienced and reported by people living with HIV accessing routine in-person outpatient HIV care in London UK. Our results are the first known to report that a sample of adults living with well controlled HIV in the UK, of any age, experience disability that is multi-dimensional and episodic in nature, using generic and HIV specific measures of disability. Disability is a universal human experience whereby individuals can be positioned on a continuum of functioning from no disability (full functioning) to complete disability (lack of function), and either currently experience or may be vulnerable to experiencing disability over the course of life [44]. Diagnosis of signs and symptoms are essential, but what most often matters is what a person can, or cannot do, in their daily life [44]. As such, our results might enable researchers, clinicians, policy makers, and people living with HIV, to better understand the nature and extent of disability among people living with HIV in the UK, consider disability-inclusive approaches to HIV care, and promote future research and national census data that includes functioning and disability measurement tools.

Prevalence of severe disability among this sample of people living with HIV was 39.5%. There are no national or international comparisons using the EADD as a measure of disability among people living with HIV. The sample of people living with HIV in this study, who were mostly of working age, economically active, and living in London, had higher estimated disability prevalence than the UK general population measured by the EADD which was estimated to be 22% [66]. Estimations are known to vary in the UK general population by age group and location; 19% of working-age adults, 45% of state pension age adults, and 15% of people living in London [66]. Limited inferences can be drawn from this data as our study did not include a control group, and no previous disability estimates exist among people living with HIV in the UK. Future research should consider matched HIV-negative control groups, to evaluate group differences, when disability prevalence is measured with EADD. Our study provides important initial estimations of severe disability among people living with HIV accessing in-person outpatient HIV clinics in the UK, for comparison to the UK general population, whereby people who experience severe disability may benefit from rehabilitation [27, 81, 82].

Prevalence of moderate disability among people living with HIV was 70.5%. This estimate was higher than disability prevalence similarly measured with ≥2 on WHODAS scores, among people living with HIV on long-term antiretroviral therapy in KwaZulu-Natal and Gauteng province in South Africa, estimated to be 35.5% and 51.9% respectively [27, 28]. Higher estimated prevalence of disability in our study measured by the WHODAS compared to South Africa, may be driven by participants living with HIV in our study reporting greater

**Table 4. Potential risk factors as predictors of HDQ domain severity scores.**

| Potential Risk Factors | HDQ Domain "Physical symptoms and impairments"; | HDQ Domain "Cognitive symptoms and impairments"; | HDQ Domain "Mental and emotional health symptoms and impairments"; | HDQ Domain "Uncertainty or worry about the future"; | HDQ Domain "Difficulties with day-to-day activities"; | HDQ Domain "Challenges to taking part in social and community life"; |
|---|---|---|---|---|---|---|
| | Estimated marginal mean of severity score (95% CI) | Estimated marginal mean of severity score (95% CI) | Estimated marginal mean of severity score (95% CI) | Estimated marginal mean of severity score (95% CI) | Estimated marginal mean of severity score (95% CI) | Estimated marginal mean of severity score (95% CI) |
| **Age** | | | | | | |
| <50 years | 31.26 (25.16, 37.36) | 36.62 (27.19, 46.05) | 44.86 (36.39, 53.34) | 53.87 (45.55, 62.19) | 30.26 (23.57, 36.96) | 41.41 (34.19, 48.63) |
| ≥50 years | 33.61 (27.26, 39.97) | 32.93 (23.11, 42.75) | 37.21 (28.38, 46.03) | 46.76 (38.10, 55.43) | 31.31 (24.33, 38.28) | 35.80 (28.28, 43.32) |
| Mean Difference (95% CI) | 2.35 (-2.27, 6.98) | -3.69 (-10.84, 3.46) | -7.66 **(-14.09, -1.23)** | -7.11 **(-13.41, -0.79)** | 1.05 (-4.03, 6.12) | -5.61 **(-11.08, -0.13)** |
| *P*-value | *P* = 0.319 | *P* = 0.312 | ***P* = 0.020** | ***P* = 0.027** | *P* = 0.686 | ***P* = 0.045** |
| **Gender Identity** | | | | | | |
| Man | 27.53 (21.31, 33.75) | 32.80 (23.18, 42.41) | 34.14 (25.50, 42.78) | 40.20 (31.71, 48.68) | 27.34 (20.52, 34.17) | 31.38 (24.02, 38.74) |
| Woman | 37.34 (29.27, 45.41) | 36.75 (24.28, 49.23) | 47.93 (36.71, 59.14) | 60.44 (49.43, 71.44) | 34.23 (25.37, 43.09) | 45.83 (36.28, 55.38) |
| Mean Difference (95% CI) | 9.81 **(1.22, 18.41)** | 3.96 (-9.32, 17.24) | 13.78 **(1.85, 25.72)** | 20.24 **(8.52, 31.96)** | 6.89 (-2.54, 16.32) | 14.45 **(4.28, 24.62)** |
| *P*-value | ***P* = 0.025** | *P* = 0.559 | ***P* = 0.024** | ***P* = 0.001** | *P* = 0.152 | ***P* = 0.005** |
| **Sexual Orientation** | | | | | | |
| Heterosexual | 30.96 (23.96, 37.97) | 36.74 (25.91, 47.56) | 39.16 (29.44, 48.89) | 54.19 (44.64, 63.74) | 32.42 (24.74, 40.11) | 38.07 (29.78, 46.36) |
| Lesbian/Gay/ Bisexual/Other | 33.91 (26.73, 41.08) | 32.81 (21.73, 43.90) | 42.91 (32.94, 52.87) | 46.45 (36.66, 56.23) | 29.15 (21.28, 37.02) | 39.14 (30.65, 47.63) |
| Mean Difference (95% CI) | 2.95 (-5.25, 11.14) | -3.92 (-16.59, 8.75) | 3.75 (-7.64, 15.13) | -7.74 (-18.92, 3.44) | -3.28 (-12.27, 5.72) | 1.07 (-8.63, 10.77) |
| *P*-value | *P* = 0.481 | *P* = 0.544 | *P* = 0.519 | *P* = 0.175 | *P* = 0.475 | *P* = 0.829 |
| **Ethnicity** | | | | | | |
| White | 32.78 (25.56, 38.99) | 37.89 (28.29, 47.49) | 42.01 (33.38, 50.64) | 50.90 (42.43, 59.37) | 32.04 (25.23, 38.86) | 39.80 (32.45, 47.15) |
| BAME | 32.10 (25.81, 38.38) | 31.66 (21.94, 41.37) | 40.06 (31.33, 48.79) | 49.73 (41.16, 58.30) | 29.53 (22.63, 36.43) | 37.41 (29.97, 44.85) |
| Mean Difference (95% CI) | -0.68 (-5.41, 4.04) | -6.24 (-13.54, 1.06) | -1.95 (-8.51, 4.61) | -1.17 (-7.61, 5.27) | -2.51 (-7.69, 2.67) | -2.39 (-7.98, 3.21) |
| *P*-value | *P* = 0.778 | *P* = 0.094 | *P* = 0.560 | *P* = 0.722 | *P* = 0.343 | *P* = 0.403 |
| **Late HIV Diagnosis** | | | | | | |
| No | 31.05 (24.74, 37.36) | 32.18 (22.43, 41.93) | 38.51 (29.74, 47.27) | 48.58 (39.98, 57.19) | 28.59 (21.67, 35.51) | 37.65 (30.18, 45.12) |
| Yes | 33.83 (27.88, 39.78) | 37.37 (28.18, 46.56) | 43.56 (35.30, 51.83) | 52.05 (43.94, 60.16) | 32.98 (26.45, 39.51) | 39.56 (32.52, 46.60) |
| Mean Difference (95% CI) | 2.78 (-1.29, 6.85) | 5.19 (-1.10, 11.48) | 5.06 (-0.60, 10.71) | 3.47 (-2.09, 9.02) | 4.39 (-0.08, 8.86) | 1.91 (-2.91, 6.73) |
| *P*-value | *P* = 0.181 | *P* = 0.106 | *P* = 0.080 | *P* = 0.221 | *P* = 0.054 | *P* = 0.437 |
| **Employment** | | | | | | |
| Economically active | 27.42 (20.95, 33.90) | 28.08 (18.07, 38.09) | 33.33 (24.33, 42.32) | 46.43 (37.60, 55.26) | 22.80 (15,70, 29.91) | 32.22 (24.55, 39.88) |
| Economic inactivity | 37.45 (31.11, 43.79) | 41.47 (31.68, 51.27) | 48.74 (39.94, 57.55) | 54.20 (45.56, 62.84) | 38.77 (31.81, 45.72) | 44.99 (37.49, 52.50) |
| Mean Difference (95% CI) | 10.02 **(4.51, 15.54)** | 13.39 **(4.88, 21.91)** | 15.41 **(7.76, 23.07)** | 7.77 **(0.25, 15.29)** | 15.97 **(9.92, 22.01)** | 12.78 **(6.25, 19.30)** |
| *P*-value | ***P*<0.001** | ***P* = 0.002** | ***P*<0.001** | ***P* = 0.043** | ***P*<0.001** | ***P*<0.001** |
| **Housing Situation** | | | | | | |

*(Continued)*

**Table 4.** (Continued)

| Potential Risk Factors | HDQ Domain "Physical symptoms and impairments"; | HDQ Domain "Cognitive symptoms and impairments"; | HDQ Domain "Mental and emotional health symptoms and impairments"; | HDQ Domain "Uncertainty or worry about the future"; | HDQ Domain "Difficulties with day-to-day activities"; | HDQ Domain "Challenges to taking part in social and community life"; |
|---|---|---|---|---|---|---|
| | Estimated marginal | Estimated marginal | Estimated marginal | Estimated marginal | Estimated marginal | Estimated marginal |
| | mean of severity score | mean of severity score | mean of severity score | mean of severity score | mean of severity score | mean of severity score |
| | (95% CI) | (95% CI) | (95% CI) | (95% CI) | (95% CI) | (95% CI) |
| Owner/Private or Social rent | 25.44 (22.25, 28.62) | 27.17 (22.25, 32.09) | 29.81 (25.38, 34.23) | 41.78 (37.44, 46.12) | 24.82 (21.32, 28.31) | 30.56 (26.79, 34.33) |
| No fixed abode/other | 39.44 (28.67, 50.20) | 42.38 (25.75, 59.01) | 52.26 (37.32, 67.21) | 58.85 (44.18, 73.53) | 36.76 (24.95, 48.56) | 46.65 (33.91, 59.39) |
| Mean Difference (95% CI) | 14.00 (**3.13, 24.86**) | 15.21 (-1.58, 31.99) | 22.46 (**7.37, 37.54**) | 17.07 (**2.26, 31.89**) | 11.94 (**0.02, 23.86**) | 16.09 (**3.23, 28.95**) |
| *P*-value | ***P* = 0.012** | *P* = 0.076 | ***P* = 0.004** | ***P* = 0.024** | ***P* = 0.050** | ***P* = 0.014** |
| **Education** | | | | | | |
| Degree level or above | 31.05 (25.00, 37.09) | 32.66 (23.32, 42.01) | 41.82 (33.42, 50.21) | 51.13 (42.89, 59.38) | 29.82 (23.19, 36.45) | 38.48 (31.32, 45.64) |
| Any other qualification | 33.83 (27.64, 40.01) | 36.89 (27.33, 46.44) | 40.25 (31.67, 48.84) | 49.50 (41.07, 57.93) | 31.75 (24.97, 38.54) | 38.73 (31.41, 46.05) |
| Mean Difference (95% CI) | 2.78 (-1.18, 6.74) | 4.22 (-1.90, 10.34) | -1.56 (-7.06, 3.94) | -1.63 (-7.03, 3.77) | 1.93 (-2.41, 6.28) | 0.25 (-4.44, 4.95) |
| *P*-value | *P* = 0.169 | *P* = 0.176 | *P* = 0.578 | *P* = 0.554 | *P* = 0.384 | *P* = 0.917 |
| **Care and Support** | | | | | | |
| Does not receive/provide care | 31.90 (25.66, 38.14) | 34.98 (25.34, 44.62) | 40.46 (31.79, 49.12) | 49.03 (40.52, 57.54) | 28.76 (21.91, 35.61) | 35.74 (28.36, 43.13) |
| Receives or provides care | 32.97 (26.54, 39.41) | 34.57 (24.63, 44.51) | 41.61 (32.68, 50.55) | 51.60 (42.83, 60.37) | 32.81 (25.75, 39.87) | 41.47 (33.85, 49.08) |
| Mean Difference (95% CI) | 1.07 (-4.10, 6.26) | -0.41 (-8.41, 7.58) | 1.16 (-6.03, 8.35) | 2.56 (-4.49, 9.62) | 4.05 (-1.62, 9.73) | 5.72 (-0.40, 11.85) |
| *P*-value | *P* = 0.684 | *P* = 0.919 | *P* = 0.752 | *P* = 0.476 | *P* = 0.162 | *P* = 0.067 |
| **Benefits** | | | | | | |
| Does not receive benefits | 28.59 (22.05, 35.13) | 28.41 (18.31, 38.52) | 36.90 (27.81, 45.98) | 47.05 (38.13, 55.97) | 25.49 (18.32, 32.67) | 34.34 (26.60, 42.08) |
| Receives benefits | 36.28 (29.87, 42.70) | 41.14 (31.23, 51.04) | 45.17 (36.27, 54.08) | 53.58 (44.84, 62.33) | 36.08 (29.04, 43.11) | 42.87 (35.28, 50.45) |
| Mean Difference (95% CI) | 7.70 (**1.88, 13.52**) | 12.72 (**3.73, 21.72**) | 8.27 (**0.19, 16.36**) | 6.53 (-1.40, 14.47) | 10.58 (**4.19, 16.97**) | 8.52 (**1.63, 15.41**) |
| *P*-value | ***P* = 0.010** | ***P* = 0.006** | ***P* = 0.045** | *P* = 0.107 | ***P* = 0.001** | ***P* = 0.015** |
| **Rehabilitation** | | | | | | |
| No rehab in past year | 28.33 (22.37, 34.28) | 30.30 (21.10, 39.50) | 38.29 (30.03, 46.56) | 46.16 (38.04, 54.28) | 25.96 (19.43, 32.49) | 34.23 (27.18, 41.28) |
| Received rehab in past year | 36.54 (30.20, 42.89) | 39.25 (29.45, 49.05) | 43.78 (34.97, 52.58) | 54.47 (45.82, 63.12) | 35.61 (28.65, 42.57) | 42.98 (35.47, 50.49) |
| Mean Difference (95% CI) | 8.22 (**4.03, 12.40**) | 8.95 (**2.49, 15.42**) | 5.48 (-0.33, 11.29) | 8.31 (**2.60, 14.01**) | 9.65 (**5.06, 14.24**) | 8.75 (**3.80, 13.70**) |
| *P*-value | ***P*<0.001** | ***P* = 0.007** | *P* = 0.064 | ***P* = 0.004** | ***P*<0.001** | ***P* = 0.001** |

Higher scores indicate greater mean severity of disability.

**BOLD** indicates mean difference 95% CI does not cross zero and *P*-value ≤0.05.

functional limitations at lower difficulty levels. In our study, 50.5% reported 4 or more functional limitations with mostly mild (38.8%) or moderate (33.7%) difficulty levels. In KwaZulu-Natal, 12.6% reported 4 or more functional limitations with 64.7% reporting moderate difficulty level [28]. Estimated prevalence of any level of functional limitation (scoring ≥1 on WHODAS) was 86.6% and 51.3% in the United States and South Africa respectively, with people living with HIV in the United States 9.76 times (95% CI: 4.91, 19.41) more likely to experience any level of functional limitation [29]. In our study, 79.5% reported any level of functional limitation. Caution should be applied when comparing these results from different contexts, employing varied measurement approaches. Our study provides important initial estimations of moderate disability (scoring ≥2 on WHODAS), suggesting that moderate disability is common among people living with HIV accessing routine in-person outpatient HIV care in London UK, and people with moderate disability may benefit from rehabilitation [27, 81, 82].

High-income settings frequently report higher rates of disability than Low- and Middle-income settings, reportedly due to ageing populations and higher survival rates for people with disabling conditions [38]. Additionally, between country differences can impact disability presence due to differences in perceptions, lived experiences, culture, lifestyle, economies, resources, education, access to and availability of antiretroviral therapy, and health policies [29]. The three disability measurement approaches used in our study provides a breadth of within-country data. This might enable critically comparative future evaluations with other populations across High- Middle- and Low-income settings. For example, our study provides the first known WHODAS "simple" and "complex" sum scores among people living with HIV in the UK. The WHODAS "simple" sum score (mean 9.4, SD 11.3) (median 5.0, 25-75[th] percentile: 1.0–12.0) were higher (meaning more disability) than men and women in the total population (mean 3.1, SD 5.3) comprising any age group including adults aged 75–85 years (mean 5.7, SD 7.1), people with ≥1 12-month mental disorder (mean 8.7, SD 7,7), people with any mental disorder (mean 6.3, SD 7.1), people with ≥1 chronic physical condition (mean 5.8, SD 7.0), and people with any physical condition (mean 4.3, SD 6.1) [58]. The WHODAS "complex" sum score (mean 19.6, SD 23.7), (median 10.4, 25-75[th] percentile: 2.1–25.6) were lower (meaning less disability) than people living with HIV in Canada (median 30, 25-75[th] percentile: 18–44) [51], comparable to people living with HIV in Ireland (median 12, 25-75[th] percentile: 5–24) [51], and higher than people living with HIV in KwaZulu-Natal (mean 1.6) and Gauteng province (mean 0.5, SD 8) South Africa [27, 28]. There is significant variation in the scoring and reporting of WHODAS sum scores among people living with HIV [27–29, 51, 83, 84], and careful attention should be applied to any comparison. Our approach used a range of measurements to permit collation of existing literature, plus contribute to establishing reproducible standards of approach, scoring, and reporting of disability in the context of HIV.

When measuring severity of disability, the HDQ domain of uncertainty was the most severe dimension of disability experienced in our study of adults living with HIV in London, UK, which is consistent with existing literature from the UK [35], Canada [51, 62, 78], Ireland [51], and United States [63]. Furthermore, the HDQ domains of challenges to social participation, and mental and emotional health, are within the top three most severe disability dimensions across the same four settings. Uncertainty is a unique dimension of disability within the Episodic Disability Framework [2], whereby people ageing with HIV may worry about HIV specific age-related uncertainties and the trajectory of episodic disability [85, 86]. Further exploration is warranted into the experiences of uncertainty across the life course among people living with HIV in the UK [35].

To our knowledge this study was the first to examine risk factors of disability across different measures of disability among people living with HIV in the UK. Our results indicated that

risk factors of disability can be divided into three different areas; HIV characteristics such as late diagnosis; social determinants of health including unemployment, housing insecurity, and receiving benefits; and personal factors such as age and gender.

The HIV characteristic of receiving a late HIV diagnosis resulted in an 171% increased risk of severe disability. Late HIV diagnosis is associated with negative clinical and societal consequences [87, 88], and further associated with disability among older adults [89, 90]. Our results indicated that the negative consequences of late HIV diagnosis may result in severe disability at any age, including among those who later attain undetectable viral load through antiretroviral therapy. Additionally, people with disabilities are known to be more vulnerable to acquiring HIV, are diverse in their HIV-risks, and more likely to receive a late HIV diagnosis [91–93].

Social determinants of health were risk factors across all measures of disability in our study. Receiving benefits resulted in 187% increased risk of severe disability, and economic inactivity resulted in 179% increased risk of severe disability, and 214% increased risk of moderate disability. Furthermore, economic inactivity, receiving benefits, and having no fixed abode were risk factors across physical, mental and emotional, difficulty with day-to-day activities, and challenges to social participation HDQ disability domains. Our results align with existing literature highlighting the role and importance of social determinants of health [94], whereby social determinants such as low level of socioeconomic status, stigma, and social exclusion, influence health outcomes and quality of life among people living with HIV [95–97]. Our results emphasise the role of social determents of health influencing disability experienced by people living with HIV in the UK.

Personal factors of identifying as a woman and being aged <50 years were both risk factors for HDQ disability domains mental and emotional health, uncertainty, and challenges with social participation. Identifying as a woman was also a risk factor for physical health challenges. Our results are consistent with identifying as a woman predicting disability [27, 28] and younger age negatively influencing quality of life [96] among people living with HIV. This is the first known study including trans women, to associate identifying as a woman and younger age with increased risk of mental and emotional health challenges, uncertainty, and challenges with social participation. Women living with HIV experiencing disability report more concerns with relationships and household activities [98], activity limitations associated with physical health challenges [28], and may lack necessary support networks due to experiences of abandonment from partners or families after disclosing HIV or disability status [98, 99]. Women living with HIV also experience high levels of poverty, psychological distress, and social isolation [100]. Future research is needed to better understand the intersectionality of disability and gender among people living with HIV in the UK. Psychometric evaluation of the HDQ among women living with HIV in the UK is needed [35].

Receiving rehabilitation services in the form of Physiotherapy, Occupational Therapy and Speech and Language Therapy in the past 12-months was found to be a predictor of disability in all models. While our cross-sectional methodology means it is not possible to determine direction of dependence between variables [101], we expect that disability was a likely predictor of participants accessing rehabilitation in this study. Ultimately, the directionality of dependence between variables may be nuanced and influenced by factors including health status, age, gender identity, socio-cultural context, economic environment, and geo-political landscape. This requires multi-stakeholder engagement to include disability within future longitudinal epidemiological HIV research and census data.

Results from the current study need to be interpreted in the context of limitations. Firstly, disability was measured through self-reported measurement tools, which may be susceptible to social desirability, recall inaccuracies and overestimate ability [30, 102, 103]. The use of self-reported measurements however does offer opportunities to ensure patients voices are at the

heart of healthcare models that need to adapt to the changing reality, in which people living with HIV will age with viral suppression [104]. Secondly, our study used EADD not Washington Group short set of questions as recommended by the United Nations to estimate disability prevalence [41]. Comparison of these different approaches identifies substantially different, although overlapping, groups as experiencing disability, attributed partly to the exclusion of mental health issues and partly to the higher severity threshold for inclusion in the Washington Group short set. Therefore, the EADD meets Great Britain's need for disability monitoring [37]. Thirdly, our definition of severe disability using EADD and moderate disability by scoring $\geq$2 on WHODAS [28] was arbitrary without validation. Nevertheless, the WHODAS cutoff has been used in other studies and this criterion provides opportunity for global comparison of self-reported disability among people living with HIV. We also included a continuous measure of disability (HDQ) in this study where interpretability of the HDQ scores as they relate to moderate or severe disability are unknown. Fourthly, we used convenience sampling with adults accessing in-person outpatient HIV care, therefore the sample may not be representative of people who do not attend in-person or the wider population of people living with HIV in the UK [105]. Nevertheless, our study provides initial estimations of disability prevalence enabling future requisite sample size calculations. Fifthly, with our cross-sectional methodology and no HIV-negative control group, no causal inferences could be established. Future research should consider recruiting a matched HIV-negative control group to evaluate difference. Lastly, this study was positioned within the theoretical framework of the ICF, which has been argued to perpetuate notions of normal/abnormal through the application of simplistic views of disability, thus reinforcing the pervasive belief that disabled bodies are inherently problematic and in need of intervention [106]. Our results do not intend to be oppressive, compound the double burden of stigma and discrimination associated with both HIV and disability [107], or focus on approaches to normalise disabled bodies [108]. Rather we aimed to provide a lens by which the self-reported functioning and disability experienced by people living with HIV in the UK can enhance person-centred HIV care, encourage disability assessment during routine HIV care, and address unmet need.

## Conclusion

People living with well-controlled HIV of any age, attending outpatient HIV care in London UK, experience and self-report multi-dimensional and episodic disability. Initial prevalence estimates of severe (39.5%) and moderate disability (70.5%), suggested disability is commonly experienced among adults accessing routine HIV care, with uncertainty the most severe domain of disability experienced. Risk factors of disability included gender, age, employment, housing, benefits, and late HIV diagnosis among people living with HIV in the UK. Results help to better understand the prevalence, severity, and risk factors of disability experienced among adults living with HIV, and identify areas in which to target interventions to reduce disability and improve optimal health and function beyond viral suppression for adults living with HIV.

## Supporting information

**S1 File. Demographic and HIV questionnaire.**
(PDF)

**S1 Table. WHODAS response per item.**
(DOCX)

## Author Contributions

**Conceptualization:** Darren A. Brown, Agnieszka Lewko.

**Data curation:** Darren A. Brown.

**Formal analysis:** Darren A. Brown, Philip M. Sedgwick.

**Funding acquisition:** Darren A. Brown.

**Investigation:** Darren A. Brown.

**Methodology:** Darren A. Brown, Kelly K. O'Brien, Richard Harding, Philip M. Sedgwick, Mark Nelson, Marta Boffito, Agnieszka Lewko.

**Project administration:** Darren A. Brown.

**Supervision:** Kelly K. O'Brien, Richard Harding, Philip M. Sedgwick, Mark Nelson, Marta Boffito, Agnieszka Lewko.

**Validation:** Philip M. Sedgwick.

**Writing – original draft:** Darren A. Brown.

**Writing – review & editing:** Darren A. Brown, Kelly K. O'Brien, Richard Harding, Philip M. Sedgwick, Mark Nelson, Marta Boffito, Agnieszka Lewko.

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
