## [Decision Letter · Decision Letter 0]

7 Feb 2022

PONE-D-21-28696Prevalence, severity, and risk factors of disability among adults living with HIV accessing routine outpatient HIV care in London, United Kingdom (UK): A cross-sectional self-report study.PLOS ONE

Dear Dr. Brown,

Thank you for submitting your manuscript to PLOS ONE. After careful consideration, we feel that it has merit but does not fully meet PLOS ONE’s publication criteria as it currently stands. Therefore, we invite you to submit a revised version of the manuscript that addresses the points raised during the review process.

Although the reviewers felt this was a timely piece of research, there were some concerns regarding the grammatical and English language editing. The author should also review PLOS guidelines regarding data availability, abstract construction and the need for line numbers

We look forward to receiving your revised manuscript.

Kind regards,

Elizabeth S. Mayne, M.D.

Academic Editor

PLOS ONE

Journal Requirements:

“This study/project is funded by the National Institute for Health Research (NIHR) Masters of Research (MRes) Programme. The views expressed are those of the author(s) and not necessarily those of the NIHR or the Department of Health and Social Care.”

Additional Editor Comments :

Although the reviewers felt this was a timely piece of research, there were some concerns regarding the grammatical and English language editing. The author should also review PLOS guidelines regarding data availability, abstract construction and the need for line numbers

Reviewers' comments:

Reviewer's Responses to Questions

**Comments to the Author**

1. Is the manuscript technically sound, and do the data support the conclusions?

Reviewer #1: Yes

Reviewer #2: Yes

2. Has the statistical analysis been performed appropriately and rigorously? 

Reviewer #1: Yes

Reviewer #2: Yes

3. Have the authors made all data underlying the findings in their manuscript fully available?

Reviewer #1: No

Reviewer #2: No

4. Is the manuscript presented in an intelligible fashion and written in standard English?

Reviewer #1: Yes

Reviewer #2: Yes

5. Review Comments to the Author

Reviewer #1: The study reported the prevalence and severity disability and risk factors among adults living with HIV in London, United Kingdom; in order to identify areas to target interventions, reduce disability, and optimise health and function. This study provides useful information and contributes substantially to the epidemiology of HCV in preparation of the elimination in the country. The manuscript is well prepared and informative with clear statement of objectives. Similarly, the methods section is appropriately explained and results section is prepared well with minor mistakes as indicated below. While publication is recommended, the manuscript requires minor revision.

The study mentioned that participants were on ART longer than 6 months, however they did not assess whether specific ART is associated with disability. Furthermore, although data on concurrent health conditions were collected and participants reporting living with a median of two concurrent health conditions in addition to HIV, additional data on the specific concurrent health condition would have added value to enhance the understanding of the risk factors associated with HIV and disability.

Abstract

The abstract is too long and it does not conform to PLOS one standards

Results

Page 15: Table 1: Participant characteristics

Gender identity: singular vs plural eg Man vs Women

Page 18, the authors did not follow PLOS instructions guidelines for labelling supplementary files

Page 18, the authors state that “Difficulty levels across all WHODAS items were “mild difficulty” n=333 (38.8%)…. while page 28 in the discussion they state “In our study, 50.5% reported 4 or more functional limitations with mostly mild (36.8%) or ……

General

There are no line numbers used in the document- the authors did not following PLOS instructions guidelines

Reviewer #2: The paper is very important and timely for the improvement of care for people with living with HIV. Though it's more or less exploratory, I believe has pointed a light to a potentially serious problem.

The paper can improve with added attention to editing. Also, the data behind findings requirement is not sufficiently fulfilled.

6. PLOS authors have the option to publish the peer review history of their article (what does this mean?). If published, this will include your full peer review and any attached files.

Reviewer #1: No

Reviewer #2: No

---

## [Author Response · Author response to Decision Letter 0]

5 Mar 2022

PONE-D-21-28696

Prevalence, severity, and risk factors of disability among adults living with HIV accessing routine outpatient HIV care in London, United Kingdom (UK): A cross-sectional self-report study.

Thank you to the journal and peer reviewers for your positive and constructive feedback. The authors are grateful for the time taken to review this manuscript and have commented on your feedback below. 

Reviewers Comments 

Reviewer #1: 

1. The study reported the prevalence and severity disability and risk factors among adults living with HIV in London, United Kingdom; in order to identify areas to target interventions, reduce disability, and optimise health and function. This study provides useful information and contributes substantially to the epidemiology of HCV in preparation of the elimination in the country. The manuscript is well prepared and informative with clear statement of objectives. Similarly, the methods section is appropriately explained and results section is prepared well with minor mistakes as indicated below. While publication is recommended, the manuscript requires minor revision.

RESPONSE: Thank you for the positive and constructive feedback. Our responses to the mistakes and minor revisions are below. 

2. The study mentioned that participants were on ART longer than 6 months, however they did not assess whether specific ART is associated with disability. 

RESPONSE: Thank you. We did not collect data on the types of medications (ART) participants were taking to treat HIV. The examination of different ART regimens, as potential risk factors for disability, was therefore not possible within this study. However, this is an important research consideration that could be included within future research into the disability experiences of people living with HIV. 

3. Furthermore, although data on concurrent health conditions were collected and participants reporting living with a median of two concurrent health conditions in addition to HIV, additional data on the specific concurrent health condition would have added value to enhance the understanding of the risk factors associated with HIV and disability.

RESPONSE: Thank you. We collected data on the number of concurrent health conditions participants were living with in addition to HIV. However, we did not collect data on the types of concurrent health conditions people were living with. Therefore it was not possible to report on the types of concurrent conditions people were living with. We agree that this additional data would have added value, and will be considered in future research into the disability experiences of people living with HIV. 

4. The abstract is too long and it does not conform to PLOS one standards

RESPONSE: Thank you. We have amended the abstract length and it now conforms with PLOS ONE standards. 

5. Page 15: Table 1: Participant characteristics

Gender identity: singular vs plural eg Man vs Women

RESPONSE: Thank you. We have amended the table to ensure that participant characteristics are singular for both Man and Woman. 

6. Page 18, the authors did not follow PLOS instructions guidelines for labelling supplementary files

RESPONSE: Thank you. We have amended the labelling of supplementary files to follow PLOS ONE instruction guidelines. The sentence has been updated to “The WHODAS simple scores in response to all 12-items are shown in S1 Table” (page 18, line 507). We have also included an additional supplementary file (S1 File) “Demographic and HIV questionnaire”, which is cited in the manuscript (page 10, line 333). 

7. Page 18, the authors state that “Difficulty levels across all WHODAS items were “mild difficulty” n=333 (38.8%)…. while page 28 in the discussion they state “In our study, 50.5% reported 4 or more functional limitations with mostly mild (36.8%) or ……

RESPONSE: Thank you. We appreciate identification of this typo. We have updated the discussion (page 28, line 647) to say the correct value 38.8%.

8. There are no line numbers used in the document- the authors did not following PLOS instructions guidelines

RESPONSE: Thank you. We have amended the document to include line numbers, and follow PLOS ONE instruction guidelines. 

Reviewer #2: 

1. The paper is very important and timely for the improvement of care for people with living with HIV. Though it's more or less exploratory, I believe has pointed a light to a potentially serious problem.

RESPONSE: Thank you for the positive feedback. 

2. The paper can improve with added attention to editing. Also, the data behind findings requirement is not sufficiently fulfilled.

RESPONSE: Thank you. We have amended the paper in response to the journal requirements and reviewer responses. We have uploaded our study’s minimal underlying data set to the public repository DRYAD. The data can be accessed on the URL and DOI below. We have updated the cover letter to include the links to the study’s minimal underlying data set. 

https://datadryad.org/stash/share/5o5ixIjJZbIjsqO9363pRnIPIg7LVUngZTYnEyX7JUc

doi:10.5061/dryad.qnk98sfjn

---

## [Decision Letter · Decision Letter 1]

6 Apr 2022

Prevalence, severity, and risk factors of disability among adults living with HIV accessing routine outpatient HIV care in London, United Kingdom (UK): A cross-sectional self-report study.

PONE-D-21-28696R1

Dear Dr. Brown,

We’re pleased to inform you that your manuscript has been judged scientifically suitable for publication and will be formally accepted for publication once it meets all outstanding technical requirements.

Kind regards,

Elizabeth S. Mayne, M.D.

Academic Editor

PLOS ONE

Additional Editor Comments (optional):

Reviewers' comments:

Reviewer's Responses to Questions

**Comments to the Author**

1. If the authors have adequately addressed your comments raised in a previous round of review and you feel that this manuscript is now acceptable for publication, you may indicate that here to bypass the “Comments to the Author” section, enter your conflict of interest statement in the “Confidential to Editor” section, and submit your "Accept" recommendation.

Reviewer #1: All comments have been addressed

Reviewer #2: (No Response)

2. Is the manuscript technically sound, and do the data support the conclusions?

Reviewer #1: Yes

Reviewer #2: Yes

3. Has the statistical analysis been performed appropriately and rigorously? 

Reviewer #1: Yes

Reviewer #2: Yes

4. Have the authors made all data underlying the findings in their manuscript fully available?

Reviewer #1: Yes

Reviewer #2: Yes

5. Is the manuscript presented in an intelligible fashion and written in standard English?

Reviewer #1: Yes

Reviewer #2: Yes

6. Review Comments to the Author

Reviewer #1: The authors have addressed all the comments according to my satisfaction. It was my pleasure to review this manuscripts

Reviewer #2: (No Response)

7. PLOS authors have the option to publish the peer review history of their article (what does this mean?). If published, this will include your full peer review and any attached files.

Reviewer #1: No

Reviewer #2: No

---

## [Editor Report · Acceptance letter]

3 May 2022

PONE-D-21-28696R1 

Prevalence, severity, and risk factors of disability among adults living with HIV accessing routine outpatient HIV care in London, United Kingdom (UK): a cross-sectional self-report study

Dear Dr. Brown:

I'm pleased to inform you that your manuscript has been deemed suitable for publication in PLOS ONE. Congratulations! Your manuscript is now with our production department. 

Kind regards, 

on behalf of

Dr. Elizabeth S. Mayne 

Academic Editor

PLOS ONE